# EAG Responses of Adult *Lobesia botrana* Males and Females Collected from *Vitis vinifera* and *Daphne gnidium* to Larval Host-Plant Volatiles and Sex Pheromone

**DOI:** 10.3390/insects10090281

**Published:** 2019-09-02

**Authors:** Alicia Pérez-Aparicio, Luis M. Torres-Vila, César Gemeno

**Affiliations:** 1Department of Crop and Forest Sciences, University of Lleida, Av. Alcalde Rovira Roure 191, 25198 Lleida, Spain; 2Servicio de Sanidad Vegetal, Consejería de Medio Ambiente y Rural PAyT, Junta de Extremadura, Avda. Luis Ramallo s/n, 06800 Mérida, Badajoz, Spain

**Keywords:** electroantennogram, host plant, volatiles, Tortricidae, sex

## Abstract

We analysed electroantennogram (EAG) responses of male and female adults of the European grapevine moth *Lobesia botrana* (Denis et Schiffermüller) (Lepidoptera: Tortricidae) collected as larvae from grapevine (*Vitis vinifera* L.) and flax-leaved daphne (*Daphne gnidium* L.). The host-plant odorants tested were either *V. vinifera*-specific [1-octen-3-ol, (*E*)-β-farnesene, (*E*)-4,8-dimethyl-1,3,7-nonatriene], *D. gnidium*-specific (2-ethyl-hexan-1-ol, benzothiazole, linalool-oxide, ethyl benzanoate), or were shared by both host-plants (linalool, methyl salicylate). Sex pheromone compounds were also tested. The male response to the major pheromone component (*E*7,*Z*9-12:Ac) was higher than to any other stimuli, whereas the response to the minor pheromone components (*E*7,*Z*9-12:OH and *Z*9-12:Ac) was not different from the response to the plant odorants. The female response to pheromone was lower or not different from that to plant odorants. Methyl salicylate elicited a higher response in females and (*E*)-β-farnesene elicited a higher response than several other plant odorants in both sexes. Non-significant interactions between host-plant odorant and sex indicated an absence of sex specialization for host-plant volatile detection. The lack of a significant interaction between plant volatiles and larval host-plants suggested that there was no specialization for plant-volatile detection between *V. vinifera* and *D. gnidium* individuals.

## 1. Introduction

The European grapevine moth, *Lobesia botrana* (Denis et Schiffermüller) (Lepidoptera: Tortricidae), is one of the most important pest species affecting vines (*Vitis vinifera* L.) and is responsible for severe economic losses in the vineyards of the Palaearctic region [1]. *L. botrana* has colonized areas of central Africa, western Asia and more recently the Americas (Chile, Argentina and the USA), making it a global grapevine pest [1,2]. Conventional control relies heavily on pesticide use, but the practice of using alternative environmentally-friendly methods has increased in recent decades. The use of semiochemicals—mainly sex pheromones and plant volatiles—in pest control is a promising tool [3]. Mating disruption (MD) is a very effective way to control insect pests, however it is rarely effective under high population densities and needs to be complemented with insecticide treatments in an Integrated Pest Management framework [1]. The success of MD in the control of *L. botrana* has stimulated research on the response of males to its sex pheromone [4,5,6,7]. However, there is growing evidence that females detect and change their calling behaviour in response to conspecific pheromone (i.e., pheromone ‘autodetection’) [8], and that this may play a role in MD [9,10]. El-Sayed and Suckling [9] determined that exposure to sex pheromone did not alter the behaviour of *L. botrana* females, but Harari et al. [10] observed a reduction in the calling behaviour and mating success of pheromone-exposed females. A comparative electrophysiological study with sex pheromone in both sexes could help explain some of these inconsistencies.

Sex bias in olfactory studies is not limited to sex pheromone and comprises plant-volatile stimuli too. Plant volatiles attract insects to their feeding, mating and oviposition sites, and help them avoid non-hosts [11,12]. Many studies on host selection in insects with non-dispersive larval stages have mainly centred on females because they are the ones that lay the eggs [13,14,15,16], while males have received less attention. Although *L. botrana* develops mainly on *V. vinifera*, it remains polyphagous and may perform better on other wild or cultivated hosts that contribute with a higher nutritional value, decrease natural predator pressure or enhance larval installation and survival [17,18]. Among these alternative hosts, the flax-leaved daphne, *Daphne gnidium* L.*,* is especially important as this thymelaeaceous, evergreen and sclerophylous shrub is considered the putative wild host of *L. botrana* [19,20]. Tasin et al. [14,16] investigated antennal and behavioural responses of *L. botrana* females to *V. vinifera*’s volatiles and compared them to the responses elicited by *D. gnidium* chemical cues. They reported that *L. botrana* females respond both to common and specific compounds of each host and to their mixture. Still, females were more attracted to the *D. gnidium*’s than to *V. vinifera*’s complete blend [16]. Some studies have explored male *L. botrana* responses to plant volatiles [21,22,23], but few have compared male and female EAG responses [24], and none, that we know, has compared individuals collected from different host plants.

In the present study, we compare electroantennogram (EAG) responses of males and females of *L. botrana* collected as larvae from *V. vinifera* and *D. gnidium* to plant volatiles from each host. We hypothesize that the response to these compounds may somehow reflect a preference for *L. botrana*’s more suited host, shedding light on the occurrence of an evolutionary host shift whereby the new moth–host association is either linked to adaptive changes or to more recent host-selection processes. In addition, we explore sex differences in the response to pheromone and plant stimuli.

## 2. Materials and Methods

### 2.1. Moth Strains and Tested Individuals

Larvae were collected from *V. vinifera* and *D. gnidium* in several locations in Extremadura, SW Spain, in 2014 (Table 1). These locations, and their agroecological contexts, were considered adequate to prevent or minimize gene flow between populations. For instance, the distance between Jaraicejo daphne stands (TJ population) and the nearest vineyards was about 12 km, and daphne plants were unknown in the Guareña vineyards (VG1 and VG2 populations). Insects were collected during May in the larval stage (mostly 4th–5th instar larvae) on grapevine inflorescences or daphne shoots. More than 50 larvae were collected per population in either commercial vineyards (>10 ha) or daphne stands (100 plants scattered over >1 ha). Larval development was completed in the laboratory at 25 ± 1 °C and 60% ± 10% RH, under a long-day photoperiod (16L:8D) on a semi-synthetic diet [25] to discard parasitized or diseased larvae and to produce similar-sized adults irrespective of population origin and host plant. Adults obtained from each strain were used as parents to produce virgin F1 adults for EAG tests as follows. Pairs of 2–3-day-old adults were caged in 22 mL clear plastic containers with water ad libitum, and green grapevine berries (vine strains) or daphne leaves (daphne strains) added into each container to promote egg laying. Containers and plant material with the eggs attached were transferred before hatching to rearing containers provided with the same semi-synthetic diet, and pupae were delivered to the University of Lleida and then sexed and kept in separated environmental chambers under a 18L:6D photoregime at 23 ± 1°C. Adults were fed 10% sucrose solution and the EAG tests were performed 2–4 days after emergence. 

### 2.2. Test Stimuli

We selected a set of behaviourally active volatile compounds from *V. vinifera* and *D. gnidium* [14,16] (Table 2). The components of the sex pheromone of *L. botrana* [6] [(7*E*,9*Z*)-dodeca-7,9-dienyl acetate (*E*7,*Z*9-12:Ac), (7*E*,9*Z*)-dodeca-7,9-dienol (*E*7,*Z*9-12:OH), and (*Z*)-9-dodecenyl acetate (*Z*9-12:Ac)], with an isomeric purity >93% (Pherobank, Wageningen, The Netherlands), were also tested. Chemicals were dissolved in GC-grade *n*-hexane (Sigma-Aldrich, Madrid, Spain) and loaded in 10 µL aliquots of 10 ng/µL of pheromone compounds or 10 µg/µL of plant compounds onto *n*-hexane pre-cleaned filter paper pieces (0.5 × 1 cm, Whatman #1, Sigma-Aldrich, Madrid, Spain). Filter papers formulated with solvent (*n*-hexane) alone were a control for solvent and puffing effects. The filter paper pieces were inserted into Pasteur pipettes five minutes after stimulus load, and the stimulus pipettes were stored in glass test tubes sealed with polytetrafluoroethylene (PTFE)-coated screw caps until used. New stimuli cartridges were prepared each day, and a given stimulus cartridge was used for no more than 12 stimulations. Test tubes for keeping stimulus pipettes were rinsed with acetone and heated at 250 °C overnight before being reused.

### 2.3. Electroantennograms

Moths were immobilized with CO_2_ for 10 s and were restrained in a handcrafted poly (methyl methacrylate) insect holder. To reduce antennal movement, the head was gently squeezed with forceps—which did not affect EAG responses—and the terminal antennal segments were excised to facilitate electric contact. Electrodes were assembled by inserting gold filaments within drawn-glass capillary tubes containing physiological saline solution (0.2M KCl). The ground electrode was inserted in the head through the mouthparts and the recording electrode contacted the tip of one antenna. The signal from the recording electrode was pre-amplified (10× gain, Universal single ended probe, Syntech, Buchenbach, Germany), high-pass filtered at 0.1 Hz, and digitalized (IDAC-4, Syntech, Buchenbach, Germany). Air flows were generated by two diaphragm aquarium pumps connected to a 3-way solenoid valve (CS-55, Syntech, Buchenbach, Germany). A 0.5 L/min flow of charcoal-filtered and humidified air blew continuously over the insect preparation through a 5 mm internal-diameter stainless steel tube placed 15–20 mm from the preparation. The stimulus pipette tip was inserted through a hole on the wall of the continuous flow tube, 110 mm from the exhaust end. A 0.5 s-long 0.2 L/min charcoal-filtered (but not-humidified) air puff was passed through the pipette to stimulate the preparation. The flow of continuous humid air decreased by 0.2 L/min during stimulation. The air around the preparation was constantly exhausted to minimize contamination. One antenna per insect was stimulated with all test compounds presented in randomized order, with a time interval of at least 60 s between puffs.

### 2.4. Statistical Analyses

The response to *n*-hexane was subtracted from the response to the test compounds in the same antenna as a control for antennal responses to physical and solvent stimulation. In the rare cases where the subtraction resulted in a negative value, a zero was used. Data were transformed [log (x + 0.1)] when needed to improve model fit. Data were analysed with linear models [lm()] using R software [26]. Models testing the effects of sex, odorant stimulus, larval host-plant and their interactions were compared with ANOVAs, preferring the simplest model that was not significantly different from the next complex one. Six individuals of each sex and host-plant were analysed. Pair-wise comparisons among treatment means were performed with estimated marginal means using the emmeans package. Raw data, R codes and selected R outputs are available online (https://repositori.udl.cat/handle/10459.1/65163).

## 3. Results

All compounds elicited measurable EAG traces that varied between −0.004 and −3.71 mV after *n*-hexane subtraction (Figure 1, Appendix A). Sex, but not host plant, had a significant effect on the response to *n*-hexane, which elicited an average response of −0.17 ± 0.05 mV and −0.62 ± 0.09 mV in females and males, respectively. The model that best explained EAG responses with a dataset containing all the odorant stimuli (pheromone and plant) included as factors (a) larval host-plant, (b) odorant, (c) sex, and (d) odorant * sex interaction (Table 3A). Thus, a substantial part of the variance in the model (i.e., a relatively high sum of squares) was associated with the interaction between sex and odorant, which resulted from the much higher male than female response to pheromone compounds (Figure 1). The next important factor contributing to the model was the odorant, whereas the contributions of sex and larval host-plant, although significant, were inferior, as indicated by their relatively smaller sums of squares (Table 3A). Due to the small host-plant effect exhibited in Figure 1, the individuals from the two host plants are mixed, but in Appendix A they are shown separately.

As the odorant * sex interaction was significant, we performed pairwise comparisons among odorants within each sex, and between sexes within each odorant (Figure 1, Appendix A). In males, the strongest response was to the major pheromone compound (*E*7,*Z*9-12:Ac), followed by the minor pheromone acetate (*Z*9-12:Ac), which in turn was not different from (*E*)-β-farnesene. In addition, all plant compounds (except 1-octen-3-ol and linalool) and the minor pheromone alcohol (*E*7,*Z*9-12:OH) produced lower responses than (*E*)-β-farnesene (Figure 1, Appendix A). In females, the response to (*E*)-β-farnesene was larger than to any pheromone compound and about 2 to 4 times higher than to the other plant stimuli. The response to the major pheromone compound in females was not different than the response to the plant compounds 2-ethyl-1-hexanol, ethyl benzanoate and linalool oxide. In further pairwise comparisons, sex was compared within each odorant and, except for the expected higher response of males to the sex pheromone stimuli, the only difference between sexes was a higher female response to methyl salicylate (Figure 1, Appendix A). Regarding the significant effect of the larval host-plant, we found that EAG amplitude was larger in the individuals from *V. vinifera* than in those from *D. gnidium* (Appendix A). Considering the different biological nature of sex pheromones and plant odorants, and the large odorant * sex interaction, the original dataset was split into two smaller subsets, one containing only pheromone stimuli (*pheromone dataset*) and another containing only plant stimuli (*plant dataset*), and each subset was analysed independently following the same ANOVA procedure as with the complete dataset. The model that best explained the *plant dataset* included the factors (a) larval host-plant, (b) plant odorant and (c) sex, without any significant interaction among them (Table 3B). Here, as with the entire dataset, the largest effect was due to plant odorants, followed by larval host-plant and sex. With the *plant dataset*, the response to (*E*)-β-farnesene was higher than the response to any other compound, while the lowest responses were to 2-ethyl-1-hexanol and ethyl benzanoate (Appendix A). EAG amplitude was larger in females than in males and, here again, *V. vinifera* individuals produced larger EAG responses than *D. gnidium* individuals (Appendix A).

In the analysis of the *pheromone dataset*, the preferred model included the factors (a) pheromone compound, (b) sex and (c) their interaction (Table 3C). The interaction was significant because females responded similarly to the three pheromone compounds whereas males responded differently to each one of them (Appendix A).

## 4. Discussion

EAG responses to the three pheromone compounds were significantly larger in males than in females as previously reported for the major pheromone compound [24]. This is common to most moth species and it is probably related to the presence of a larger number of sex-pheromone olfactory receptor neurons (ORNs) in males than in females [27]. In males, the EAG response ratios of *Z*9-12:Ac and *E*7,*Z*9-12:OH, relative to *E*7,*Z*9-12:Ac, were 0.25 ± 0.05 and 0.06 ± 0.02 (mean ± SEM), respectively. These ratios correspond roughly with the proportion of the three pheromone components in the female blend (1:0.2:0.01, reviewed in [6]). Possibly, the proportion of ORNs tuned to each of the three pheromone compounds is relatively similar to the proportion of these compounds in the blend, as occurs in other moth species [28] and references therein. A single-cell electrophysiology study shows that 50% of the *L. botrana* male ORNs respond to *E*7,*Z*9-12:Ac [29], which supports this hypothesis. 

Female moth response to conspecific sex pheromone remains relatively unexplored [8]. In *L. botrana*, there is behavioural evidence for autodetection of the major pheromone compound [10]. In our study, none of the pheromone compounds elicited significantly higher responses than *n-*hexane in females (data not shown, *p* > 0.05). De Cristofaro et al. [29] found that 7% of the *L. botrana* female ORNs respond to *E*7,*Z*9-12:Ac. Whether these cells are specific to the major compound or respond to other pheromone compounds, or whether there are ORNs specifically tuned to each pheromone compound in females, needs to be determined.

Analysis of the *plant dataset* revealed larger female than male EAG amplitudes, that is, regardless of the plant stimulus or host-plant population, the EAG amplitude was larger in females than in males. Although the effect of sex was significant, the plant-odorant * sex interaction was not, which means that regardless of larger EAG female responses, the relative response to the panel of plant odorants was not different between sexes. Larger female than male overall EAG amplitude to plant volatiles could result from females having more plant-sensitive ORNs than males. In several moth species, the effect of sex on EAG responses to plant compounds depends largely on the compounds tested [24,30,31,32], while in other cases the responses are consistently larger in one sex than in the other, as in *Hyles lineata* F. (Lepidoptera: Sphingidae) [33] or *Agrotis segetum* Denis et Schiffermüller [34]. In general, reported sex difference in EAG amplitude to plant compounds is relatively small, i.e., less than 1-fold, as we found in *L. botrana*, which suggests that plant detection is important to both sexes, even though males do not need to locate the host plant to lay eggs.

Methyl salicylate produced significantly larger EAG responses in females than in males (1.89 times). Higher female than male EAG responses to this compound have also been reported in another tortricid moth, *Cydia strobilella* L. [35] (1.46 times higher), and in a sphingid moth [33] (2.63 times higher). In *Cydia pomonella* L., the response to methyl salicylate also appears to be larger in females than in males [32]. Methyl salicylate is a common plant compound involved in stress signalling [36], and it attracts natural predators [37]. This odorant discourages female *Mamestra brassicae* L. oviposition [38]. Both male and female *M. brassicae* show similar EAG responses to methyl salicylate [39], and both have an olfactory receptor neuron type that is very specific and sensitive to this compound [38]. An odorant receptor protein (EpOR1) highly sensitive to methyl salicylate has been identified in the tortricid moth *Epiphyas postvittana* Walker and has a similar level of expression in both sexes [40]. Methyl salicylate is also one of the most active compounds eliciting locomotor behaviour of *L. botrana* larvae in a servosphere [41]. Thus, this compound is probably important for *L. botrana*, and further behavioural studies concerning it are warranted.

(*E*)-β-farnesene elicited the strongest response in both sexes, despite having the lowest vapour pressures of all the plant compounds tested. (*E*)-β-farnesene is one of the key volatiles that conform the most attractive *V. vinifera* volatile blend to *L. botrana* mated females, but has not been detected in volatiles collected from *D. gnidium* [16]. Relatively specific ORNs to this compound have been described in sensilla trichodea or auricillica of males and females of the tortricid moths *C. pomonella* [42] and *Grapholita molesta* Busk [43]. Vitagliano et al. [24] reported larger EAG responses to (*E*)-β-farnesene in *L. botrana* males than in females. The difference with our study could be due to the use of different puff stimuli for the standardization of EAG responses (*Z*-3-hexenol in their study, *n*-hexane in ours).

Although global EAG amplitude to plant compounds was larger in the individuals collected from *V. vinifera* than in those from *D. gnidium*, relative sensitivity to plant compounds did not differ between the two groups, as indicated by the lack of a significant host * odorant interaction, which suggests that odour blends containing mixtures of these compounds should be perceived similarly by the two host-plant insect groups. Behavioural responses to plant odours require brain integration of antennal input from each individual odorant in the odour blend [44]. Therefore, a lack of statistical differences in odour discrimination at the antennal level does not imply that insects from each host would show similar preference for the two host-specific odour blends. Differences in plant preference could still occur if there are not differences at the EAG level.

The set of plant odorants tested in our study (4 *Vitis*-specific, 3 *Daphne*-specific and 2 common to both hosts) was chosen based on behavioural activity, host specificity and commercial availability. Other plant odorants like (*E*)-β-caryophyllene, which is one of the essential components in the *V. vinifera* blend [45], or the volatiles of tansy (*Tanacetum vulgare* L.), the flowers of which attract *L. botrana* adults, although it is not a suitable larval host [46], were not included in our study. It is possible that EAG differences among the two host groups may show with other plant compounds. Yet, our study provides a first assessment of the effect of larval host-plants on *L. botrana* olfaction, and the lack of host-odour specificity suggests that if an evolutionary host-shift from *D. gnidium* to *V. vinifera* has taken place in the past [20], it has not been accompanied by a significant modification of the moth sensory system. Reproductive isolation, if any, has not substantially modified sex pheromone detection either.

## 5. Conclusions

Our study reveals no antennal discrimination by male and female *L. botrana* adults collected as larvae from *D. gnidium* or *V. vinifera* to a set of volatiles released by both hosts or specific to each one of them. This finding suggests that males and females have a similar peripheral odour detection system to this set of plant volatiles and argues against host-plant volatile-detection specificity between individuals collected in one host or the other. Reproductive isolation, if any, has not substantially modified sex pheromone detection either. Two plant volatiles—methyl salicylate and (*E*)-β-farnesene—have shown significant antennal responses and could have a relevant role in behaviour. Small, but significant pheromone response in females grant future SSR studies to accept or discard the pheromone autodetection hypothesis, which could be relevant in MD control strategies.

## Figures and Tables

**Figure 1 insects-10-00281-f001:**
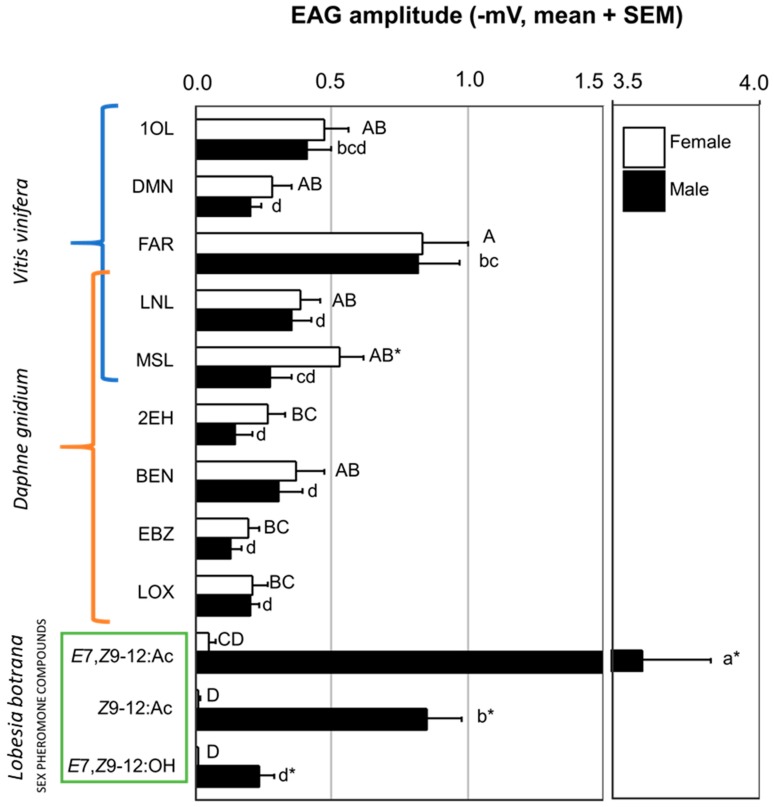
Electroantennogram responses of *L. botrana* to host plant odorants and individual pheromone compounds. Individuals from the two host plants (*V. vinifera* and *D. gnidium*) are mixed. Different letters indicate significant differences among stimuli within each sex (capital letters for females, lowercase for males), and (*) indicates significant difference between sexes (Tuckey’s pairwise test after ANOVA [Table 3A], *p* < 0.05). Plant compounds: 1OL, 1-octen-3-ol; 2EH, 2-ethyhexan-1-ol; BEN, benzothiazole; DMN, (*E*)-4,8-dimethyl-1,3,7-nonatriene; EBZ, ethyl benzanoate; FAR, (*E*)-β-farnesene; LNL, linalool; LOX, linalool oxide; MSL, methyl salicylate.

**Table 1 insects-10-00281-t001:** Moth larval host-plants and field collection sites.

Code	Host Plant	Collection Date	Municipality	Site	WGS84 Coordinates
VA	*V. vinifera*	14 May 2014	Arroyo de San Serván	El Calvario	38.853554, −6.442819
VG1	*V. vinifera*	22 May 2014	Guareña	Sartenillas	38.887461, −6.148765
VG2	*V. vinifera*	22 May 2014	Guareña	Pozo Calero	38.895705, −6.137196
TA	*D. gnidium*	14 May 2014	Arroyo de San Serván	Dehesa Grajera	38.860016, −6.436377
TJ	*D. gnidium*	20 May 2014	Jaraicejo	La Sarna	39.668019, −5.794076
TM	*D. gnidium*	20 May 2014	Madroñera	Dehesa de la Solana	39.442581, −5.791424

**Table 2 insects-10-00281-t002:** Plant stimuli for electroantennogram (EAG) tests.

Compound	Abbr	Host	CAS Number	Product Number (Sigma Aldrich)	Lot Number	Purity ^a^ (≥ %)
1-Octen-3-ol	1OL	*V. vinifera*	3391-86-4	O5284	PR 03904AQ	98
(*E*)-4,8-Dimethyl-1,3,7-nonatriene	DMN	*V. vinifera*	19945-61-0	^d^		
(*E*)-β-farnesene	FAR	*V. vinifera*	18794-84-8	73492		90
2-Ethyl-1-hexanol	2EH	*D. gnidium*	104-76-7	04050	BCBJ9176V	99
Benzothiazole	BEN	*D. gnidium*	95-16-9	W325600	STBC5100V	96
Ethyl benzanoate	EBZ	*D. gnidium*	93-89-0	W242209	STBC8296V	99
Linalool oxide ^b^	LOX	*D. gnidium*	60047-17-8	62141	BCBM5843V	97
Linalool ^c^	LOL	Both	78-70-6	L2602	STBC9155V	97
Methyl salicylate	MSL	Both	119-36-8	^d^		

^a^ As indicated by manufacturer. ^b^ Mixture of isomers.^c^ Racemic.^d^ Present from Ashraf El-Sayed, New Zealand.

**Table 3 insects-10-00281-t003:** ANOVA results using different data sets. (A) All the odorant stimuli (pheromone and plant compounds), (B) plant compounds only, (C) pheromone compounds only.

**A. Pheromone and Plant Stimuli**
**Source of Variation**	**Df**	**Sum Sq.**	**Mean Sq.**	**F**	**Pr(>F)**
Larval Host-Plant	1	5.2	5.20	15.1	0.00013
Odorant	11	53.1	4.83	14.1	<2 × 10^−16^
Sex	1	7.0	7.03	20.4	9.6 × 10^−6^
Odorant * Sex	11	83.5	7.59	22.1	<2 × 10^−16^
Residuals	239	82.1	0.34		
**B. Plant Stimuli**
**Source of Variation**	**Df**	**Sum Sq.**	**Mean Sq.**	**F**	**Pr(>F)**
Larval Host-Plant	1	5.3	5.33	13.78	0.00027
Plant Odorant	8	26.7	3.34	8.64	5.3 × 10^−10^
Sex	1	3.1	3.15	8.13	0.00483
Residuals	187	72.3	0.39		
**C. Pheromone Stimuli**
**Source of Variation**	**Df**	**Sum Sq.**	**Mean Sq.**	**F**	**Pr(>F)**
Pheromone Compound	2	26.3	13.2	66.3	6.3 × 10^−16^
Sex	1	70.1	70.1	353.6	<2 × 10^−16^
Pheromone Compound * Sex	2	15.1	7.5	38.0	2.1 × 10^−11^
Residuals	60	11.9	0.2

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
