# Peer review of "EAG Responses of Adult Lobesia botrana Males and Females Collected from Vitis vinifera and Daphne gnidium to Larval Host-Plant Volatiles and Sex Pheromone"

_insects, 2019, doi:10.3390/insects10090281_

Round 1
Reviewer 1 Report
Line 61: The authors mention that one study already compared male and female EAG responses (Vitagliano, S. et al. 2005), and also other studies worked on the topic of EAG responses in Lobesia before. In the introduction, I was not able to identify what is new about the current study and what it adds to the already published literature. Adding this point in the introduction is necessary.
Line 133: The authors tested 6 individuals of each sex and host plant, this seems to me like a rather small sample size for EAG recordings. Were there any effects regarding the sample size, e.g. if the statistical analysis is only done for 3 individuals, do you get the same result? In other words, is the individual variation so low that 6 individuals are enough to get a complete picture of the EAG responses?
Line 141-143: Sex had an influence on the hexane response, do you have an explanation for this?
Lines 162-164: "Regarding the significant effect of the larval host-plant, we found that EAG amplitude was larger in the individuals from V. vinifera than in those from D. gnidium." Please show these data, as you also discuss this point in the discussion.
Figure 1: Does Figure 1 include a mixture of individuals collected from both plants? If so, please mention it in the figure legend.
Line 206: "In our study, female response to the pheromone compounds was very small..." Please test whether the response to pheromones in females was significantly larger than their response to hexane. Just by looking at Figure 1, this does not seem to be the case (response almost zero). If this is not the case, your data set does not support your conclusion that "This suggests that some female ORNs may be tuned to E7,Z9-12:Ac (line 208)".
Line 212: "The analysis of the plant dataset revealed larger female than male EAG amplitudes, but a lack of plant-odorant*sex interaction also indicated that the relative response to plant odorants was similar in males and females." This sentence is really confusing. Is the response of females significantly larger than the response of males or not? Please clarify. When I look at Figure 1, this seems to be only true for one compound: methyl salicylate. This compound is discussed in the next paragraph (starting from line 222). So I'm not sure about the take home message of the paragraph from line 212 -221. This needs to be changed.
Line 240: "Vitagliano et al. [24] reported larger EAG responses to (E)-β-farnesene in L. botrana males than in females." The authors did not find the same results, this should be discussed.
Line 257: "our study provides a first assessment on the effect of larval host-plant on L. botrana olfaction" If this is the main focus of your paper, you need to present a graph showing the responses of the two host strains, rather then just mentioning this in the text.
Lines 273 and 274: Supplementary Materials: The following are available online at www.mdpi.com/xxx/s1, Figure S1: title, Table S1: title, Video S1: title. Are there any supplementary material? I could not find them via the link and there is also nothing mentioned in the text.
Author Response
We have taken in consideration all the comments and made changes in the manuscript, which we detail in the following point-by-point response table attached as a PDF.

Reviewer 2 Report
Overall this is an interesting piece of work that provides a good starting platform for further research into host plant selection by the European grapevine moth. English language and style is largely fine, but does need some editing for clarification in places. The experimental design is appropriate for the study although the statistical analyses could be better explained/details could be included in the results section of the manuscript text. Detailed comments can be found below:
ABSTRACT
P1, L13-15: Consider including the common name for all study species.
INTRODUCTION
The introduction is generally well written and provides a comprehensive overview of the knowledge surrounding L. botrana host selection. It was good to see the hypothesis clearly stated. Just a few minor points:
P1, L30-32: Is there a reference to support this information? P1, L40: Include the names of the pheromone compounds. P1, L41-43: Unclear sentence – in what way do they change their behaviour? Integrate this with L43-45 to clarify. P2, L52: Change ‘which’ to ‘that’. P2, L53: Remove ‘that’. P2, L58: Provide more detail on the mixture – is it a full blend or selected chemicals?
MATERIALS AND METHODS
Some sections of the M&M require clarification or further detail as it would be difficult to replicate the experiments in their current form. Generally, the EAG method is appropriate.
P2, L78: ‘RH’ needs to be uppercase. P2, L78: Hyphenate ‘semi-synthetic’. P2, L78: Add ‘a’ between ‘on’ and ‘semi-synthetic’. P2, L82: Remove ‘were’. P2, L84: Similar semi-synthetic diet? If the diet has been modified this needs to be clearly stated with the modifications listed. P2, L86: ‘Lights off at 10:00AM’ is an unnecessary detail. P3, L94: Manufacturer of the solvent? P4, L111: Why were gold filaments used? Silver is the classic filament choice. P4, L130-131: Which version of R was used? P4, L135: Reference for emmeans? P4, L137-138: In my opinion this sentence isn’t required as this is universally acknowledged.
RESULTS
Please report statistics in the manuscript text. Example EAG recordings may be beneficial for the reader.
P5, L170: Remove ‘again’ P5, L175: Can you quantify the differences in the text or refer to a figure? Just stating that EAG responses were higher for one host plant compared to the other is quite vague.
DISCUSSION AND CONCLUSION
P7, L194: Comma in this sentence is unnecessary. P7, L204: You have already introduced the concept of autodetection in the introduction, you can probably remove the information within the parentheses. P8, L252: Change ‘was’ to ‘were’.
GENERAL COMMENTS
References 18 and 29 are not formatted correctly. Abbreviating the chemical compounds throughout the manuscript seems unnecessary, especially as the abbreviations don’t match with those frequently used in the literature (e.g. methyl salicylate = MeSA). The abbreviations could be kept in figure 1.Author Response
We have taken in consideration all the comments and made changes in the manuscript, which we detail in the following point-by-point response table attached as a PDF.

Round 2
Reviewer 1 Report
I agree to the changed version of the manuscript. The authors included all suggestions from the reviewers and the paper is in my opinion ready for publication.